# Autumn Tillage Reduces the Effect of Plant Cover on Topsoil Nitrogen Leaching

**Jorge F. Miranda-Vélez** [1,*] and **Iris Vogeler** [1,2]

1   Department of Agroecology, Aarhus University, 8830 Tjele, Denmark; iris.vogeler@agro.au.dk
2   Grass and Forage Science/Organic Agriculture, Christian-Albrecht University of Kiel, 24118 Kiel, Germany
*   Correspondence: jorge_mv@agro.au.dk

**Abstract:** Keeping cover crops to reduce nitrogen leaching often conflicts with timing tillage operations before the soil becomes un-trafficable during winter, while leaving cover crops in the field until spring raises concerns over pre-emptive competition with the following crop. Therefore, farmers may resort to tilling their fields in autumn after letting cover crops remain in the fields for only a short period of time. We explore the effects of this practice in a laboratory lysimeter setting by analyzing the leaching of nitrate from intact topsoil cores. Cores were extracted from no-till (NT) plots and plots tilled in autumn (AuT), in areas kept bare (B) and with volunteer winter rye plant cover (V) after harvest. Nitrate breakthrough curves show that V significantly reduced N leaching by 61% relative to B in NT, but did not have a significant effect in AuT. Dissection of leached cores and undisturbed reference cores indicated a significant removal of mineral N from the soil during the lysimeter experiment for all treatments except V in NT. This indicates that volunteer cover removed a crucial amount of leachable N and suggests that tillage counteracted the effect of V in AuT, likely due to a combination of reduced uptake and re-mineralization of N in cover crop residue.

**Keywords:** nitrogen; nitrogen leaching; autumn tillage; no-till; lysimeter

## 1. Introduction

Keeping plant cover between harvest and the sowing of the subsequent cash crop, i.e., cover crops, has many advantages in agriculture. Cover crops can be used as a natural control against weeds [1] and to reduce nitrogen leaching after harvest [2,3]. Additionally, cover crops are known to protect agricultural soil from erosion, improve soil structure and fertility, and stimulate carbon storage in the soil [4–6].

Concerns over pre-emptive competition over nutrients [7] and water [8], as well as poor field trafficability in early spring and autumn spreading of animal manures [9], often motivate farmers to limit the residence time of winter cover crops. Thus, cover crops successfully established after harvest may be terminated in the same autumn, depending on the farmer's judgement. In the Nordic countries, for instance, the use of post-harvest cover crops is either compulsory with area requirements depending on management (Denmark) or is encouraged and subsidized (Sweden, Norway and Finland). However, farmers are typically free to till their fields and terminate the cover crops after 20 October in Denmark and Sweden and after 1 October in Finland. Only in Norway is the associated subsidy contingent on spring termination of the cover crops [10].

Cover crops in temperate climates extend roots and take up soil N during a relatively short growth period before being killed by frost or terminated by the farmer. During this period, however, post-harvest N uptake by cover crops in temperate climates frequently represents over 20% of the applied fertilizer N, resulting in a 35% average reduction in soil mineral N in autumn and reducing nitrate leaching by 40 to 70% [11,12]. Upon cover crop termination, litter and root breakdown in the soil results in a gradual re-release of a significant fraction of the taken-up N. The timing and extent of this process is affected

by a number of factors—e.g., type of cover crop, degree of incorporation, air and soil temperature, precipitation, and termination time—and is key to the overall effect of the cover crops [13]. If a significant amount of soil N is taken up by the cover crops and re-mineralized during the following growth season, it will be available for uptake by the cash crop and less fertilizer need to be applied to the soil. Indeed, Danish regulations require a fertilizer reduction of up to 25 kg N ha$^{-1}$ following cover crops in order to utilize the N "carried over" by the cover crops [10]. However, if the time for soil N uptake by cover crops is short and/or re-mineralization of cover crop N takes place during a period of fallow with high precipitation, it will be at high risk of leaching [14,15]. These considerations, unfortunately, often conflict with farmer's habits and their concerns involving weed control (including the cover crop) and reduced biomass and grain yields on the following cash crop [10,16,17].

This study explores the effect of autumn tillage on the capacity of cover crops to reduce nitrogen leaching. We compared the effect of autumn inversion tillage (AuT) against no-till (NT) on the reduction of nitrogen leaching by volunteer winter rye as cover crops in a laboratory lysimeter setting. We expect that the effectiveness of the volunteer cover crops will be diminished by autumn tillage for two reasons: (1) shortening of the time available for the volunteers to take up N from the soil and (2) re-mineralization of nitrogen in the incorporated volunteer residue. Thus, we hypothesize that soil mobile N contents will be high in AuT regardless of cover crop treatment, and that leached N recovery from AuT cores with volunteer cover crops will be greater than in corresponding NT cores.

## 2. Materials and Methods

### 2.1. Field Operations and Sampling

Sampling took place in the CENTS long-term rotation and tillage experiment at the department of Agroecology, Aarhus University Flakkebjerg (55°19′ N, 11°23′ E) [18]. The soil is a sandy loam with 14.7% clay, 13.7% silt and 69.6% sand contents, average bulk density of 1.53 Mg m$^{-3}$ and an average organic matter content of 2% [19]. Soil pH (water suspension) at the time of sampling ranged between 7.4 and 8.3, with a mean value of 7.8. The long-term experiment consists of a randomized split-plot design with four crop rotations (R1–R4) as main plots and four levels of tillage (direct sowing, shallow harrowing, deep harrowing and inversion tillage) as sub-plots, arranged in four replicate blocks [20]. For this study, sampling was restricted to the R2 rotation, which consists of 3 years of winter barley (*Hordeum vulgare*) followed by 1 year winter rape (*Brassica napus*) and 2 years winter wheat (*Triticum aestivum*), all with straw retention after harvest [18]. The last crop before sampling was substituted with winter rye (*Secale cereale*) followed by fodder radish (*Raphanus sativus*) as a winter cover crop. Sampling was also restricted to direct sowing (NT) and inversion tillage (AuT). Finally, only three of the four experimental blocks in the long-term experiment were used for sampling and are considered here as replicate samples.

The NT treatment consisted of sowing by direct drilling using a single-disk drill (2002 to spring 2006) and later a single chisel coulter drill (autumn 2006–present), and straw retention at harvest. The AuT treatment consisted of seeding with a drag coulter seed drill, straw retention at harvest and inversion ploughing followed by rolling before sowing [20]. On the year of sampling, inversion tillage in the AuT sub-plots was carried out in late October.

In late August, glyphosate (1 L ha$^{-1}$ Roundup Bio) was applied to the NT and AuT sub-plots in the R2 rotation, killing the established mixture of cover crops, volunteers and weeds. This allowed the center portion of each sub-plot to remain bare (B), while the edges of the sub-plots were re-established exclusively with winter rye volunteers (V). Clippings carried out in the V portions of AuT showed that a growth of approximately 600 kg ha$^{-1}$ DW in aboveground winter rye biomass at the time of tillage in October.

In early December, two intact topsoil cores (Ø = 20 cm, h = 20 cm) were extracted side by side from the center (B) and edge (V) of each AuT and NT subplot. Core extraction was carried out by slowly pressing a steel cylinder into the soil with a hydraulic press,

and subsequently digging around the buried cylinder with spades to manually retrieve the cores. This resulted in two separate sets of twelve samples covering four treatment combinations (AuT/NT × B/V) per block. One set of cores was designated as reference for field conditions (Reference), while the second set of cores was designated as experimental set to undergo simulated precipitation in the laboratory (lab-rain). All soil cores were transported back from the field and put in storage at 2–4 °C on the same day as extracted, awaiting sample preparation.

Additionally, composite topsoil samples (0–20 cm depth) were taken from the core sampling areas of all tillage and cover crop treatment combinations. Representative sub-samples of loose soil were then used for total soil C and total N content analysis in a Vario Max Cube organic elemental analyzer (Elementar, Germany).

### 2.2. Sample Handling and Preparation

The soil water content of all cores, both Reference and lab-rain, was equalized before the leaching experiment. Each intact core was moved into a 5 °C temperature-controlled room and placed on a ceramic plate inside a ~40 L plastic tub and slowly saturated with a simulated soil solution ($CaCl_2$ 0.05 M) (Figure 1A). After 7 days of saturation, each core was slowly drained to a pressure of −10 hPa using the same ceramic plates and a controlled vacuum system. Upon reaching equilibrium with the vacuum system, lab-rain cores were transferred to a benchtop lysimeter for a leaching experiment. The Reference cores, in turn, were removed from the ceramic plates upon reaching equilibrium at −10 hPa, and kept lidded at 5 °C for the duration of the leaching experiment.

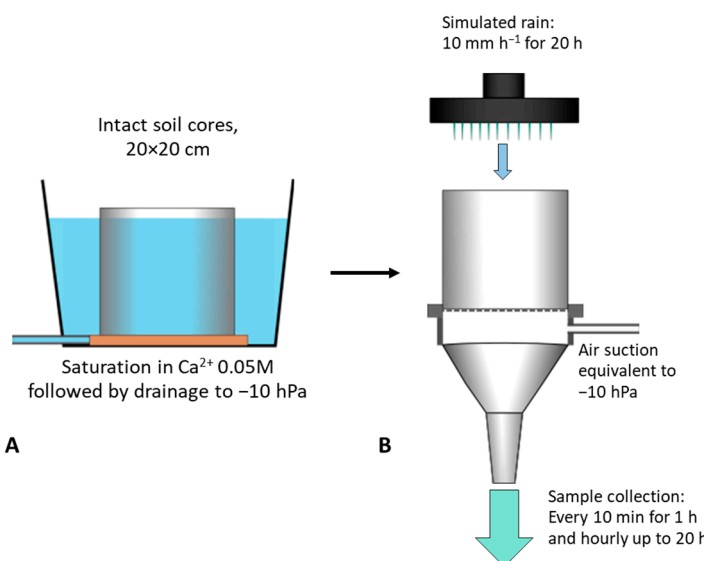

**Figure 1.** Sample preparation (**A**) and leaching experiment (**B**) setup. Preparation consisted of saturating intact soil cores in a simulated soil solution followed by drainage and equilibration at −10 hPa. In the leaching experiment, the soil cores were irrigated with simulated rain at a rate of 10 mm h$^{-1}$ for 20 h, while leachate was collected.

### 2.3. Leaching Experiment

The leaching experiment consisted of 200 mm of simulated precipitation, administered in a single course of 20 h at a constant rate of 10 mm h$^{-1}$ on individual benchtop lysimeters (Figure 1B). The lysimeters were fitted with rotating rain heads utilizing blunt needles as dripping nozzles. In order to prevent soil dispersion during the leaching experiment, simulated rainwater consisted of a weak salt solution (1.76 mg L$^{-1}$ $CaCl_2 \cdot 2H_2O$ + 3.05 mg L$^{-1}$ $MgCl_2 \cdot 6H_2O$ + 7.07 mg L$^{-1}$ NaCl), similar to that used in other leaching studies (e.g., [21,22]). The total simulated precipitation, 200 mm, corresponds approximately to the total precipitation recorded by the meteorological station at AU Flakkebjerg (211 mm) during the months

of October, November and December of 2019. For comparison, the total normal autumn precipitation (September–November, 1961–1990) in Denmark is 228 mm [23]. The leachate from each core was collected in pre-weighed plastic bottles using automatic rotating carrousels at 10-min intervals during the first hour and at 1-h intervals until termination of the experiment.

The nitrate in the collected leachate was quantified by ion chromatography using a Metrosep cation resin suppressor (Metrohm, Switzerland) and an A Supp 5 anion exchange column followed by an electric conductivity detector, with a carbonate buffer as eluent. Leachate samples were filtered using cartridge polyether sulfone (PES) filters (pore size 0.22 μm) before injection.

Following the leaching experiment, both the Reference and lab-rain soil cores were dissected in four layers, corresponding to depths of 0–5 cm, 5–10 cm, 10–15 cm and 15–20 cm, relative to the metal core casings. Each layer in each core was weighed immediately after dissection and placed in an airtight plastic bag, then stored at 2 °C awaiting sub-sampling.

Two representative soil sub-samples (~50 g) were taken from each layer in each core. The first set dried at 105 °C for 24 h for water content determination at the time of subsampling. The second set of subsamples was weighed and placed in vials for $NO_3^-$ and $NH_4^+$ (mineral N) extraction. Mineral N extraction consisted of suspending the soil samples in 0.2 L of KCl 1M solution and mixing for 30 min in a rotary shaker at 20 rpm, then filtering the supernatant using ashless paper filters. The collected extract was then frozen at −20 °C awaiting analysis. Nitrogen as nitrate and as ammonium in the extracts were quantified colorimetrically in a Seal Analytical AA500 auto-analyzer as described by Best [24] and Crooke and Simpson [25], respectively.

### 2.4. Calculations and Statistical Analysis

All data handling, visualization and analysis was carried out using R version 4.1.0 "Camp Pontanezen", released in May 2021 [26].

Total C, total N and pH measurements were analyzed using linear mixed models with tillage and cover crop treatments as main effects and field block as random effect (packages lme4 and lmerTest [27,28]).

Nitrate leaching (mg) was calculated as the product of the collected volume of leachate at each sampling time and the leachate $NO_3^-$ concentration as determined by ion chromatography. The total leached N as $NO_3^-$ ($N_{NO3}$, mg N) was calculated as the sum of the leached $NO_3^-$ over the course of the leaching experiment, multiplied by the ratio of the atomic mass of N to the molar mass of the $NO_3^-$ ion. The mass of mineral N ($N_{min}$, mg N) extracted from soil subsamples was calculated as the sum of N as $NO_3^-$ and $NH_4^+$. Total $N_{min}$ was then calculated for each dissection layer and full cores using the wet mass at dissection and the determined water content.

Differences in $N_{NO3}$ leaching were analyzed using linear mixed models, with total leached $N_{NO3}$ as response variable, tillage and plant cover as main effects and field block as random effect. Model estimates and 95% confidence intervals for leached $N_{NO3}$ were obtained from model marginal means (package emmeans [29]) and significant differences were evaluated by multiple pairwise comparison among all tillage and plant cover treatment combinations (package multcomp [30]).

The effects of tillage and plant cover on the differences between the total $N_{min}$ content of the Reference and lab-rain cores was evaluated using linear mixed models with experimental group (i.e., Reference or lab-rain), tillage and plant cover as main effects and field block as random effect. Here, all interactions including the three-way interaction were explicitly preserved in the model in order to evaluate the influence of tillage and cover crop treatments on the difference between Reference and lab-rain cores. Model estimates and 95% confidence intervals were obtained from model marginal means and significant differences between Reference and lab-rain were evaluated by multiple pairwise comparisons grouped by tillage and plant cover treatment combinations.

## 3. Results and Discussion

### 3.1. Breakthrough Curves and $N_{NO3}$ Leaching

The $NO_3^-$ breakthrough (Figure 2(top)) and cumulative $N_{NO3}$ leaching (Figure 2(bottom)) curves show different effects of cover crops on N leaching from the topsoil, depending on tillage. In the NT treatment, there is a clear difference due to plant cover in both leachate $NO_3^-$ concentrations and total $N_{NO3}$ leached over the course of the experiment. Specifically, $NO_3^-$ concentrations were lower and there was less $N_{NO3}$ leached under volunteer plant cover, compared to bare soil. In AuT, the difference between B and V was much more subtle for both $NO_3^-$ concentrations and accumulated $N_{NO3}$ leaching throughout the experiment. Additionally, NT had an overall effect on the elution of nitrate from the intact soil cores, independent of plant cover, where $NO_3^-$ peaked earlier and more sharply in NT compared to AuT. This, in turn, indicates increased preferential flow due to better-developed macropore flow pathways [31] in NT.

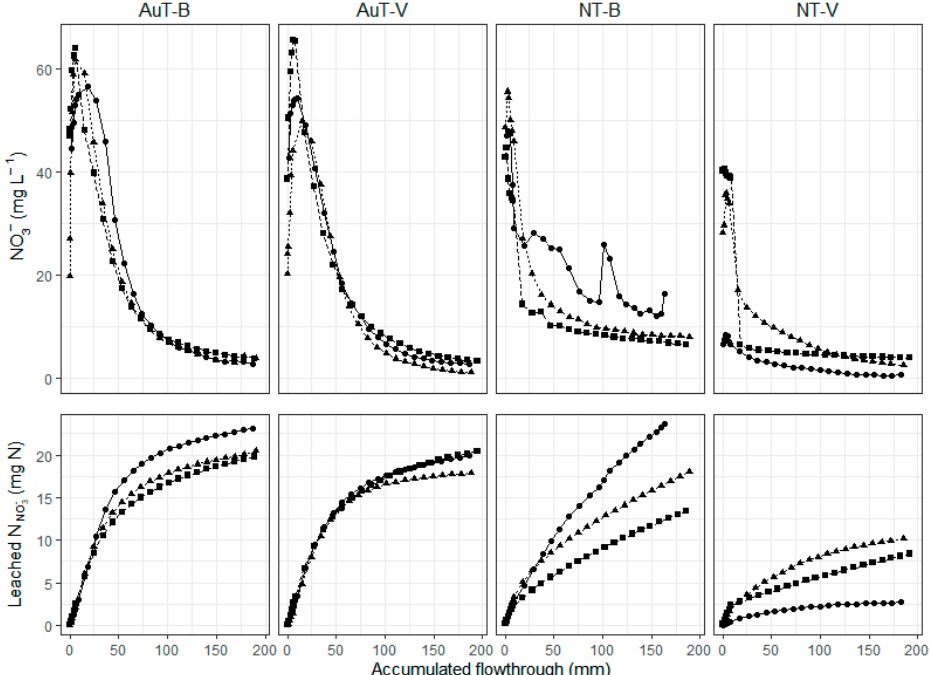

**Figure 2.** Nitrate concentration breakthrough (**top**) and accumulated leaching of $NO_3^-$ N (**bottom**) curves from intact soil cores subjected to ~200 mm of simulated rain in a laboratory lysimiter. The cores belonged to two tillage treatments, no-till (NT) and autumn inversion tillage (AuT), as well as two cover crop treatments, bare fallow (B) and winter rye volunteers (V). Symbols indicate different field experiment blocks.

Statistical analysis of total $N_{NO3}$ leaching showed significant effects by both plant cover (F = 10.98, df = 8, *p* = 0.010) and tillage (F = 14.72, df = 8, *p* = 0.005), as well as a significant interaction between plant cover and tillage (F = 5.88, df = 8, *p* = 0.041). Pairwise comparisons (Table 1) show that $N_{NO3}$ leaching under simulated precipitation was significantly lower in V compared to B in NT. The reduction corresponds to 11.3 mg N, i.e., 61% of the $N_{NO3}$ leached in the bare soil treatment of NT. In contrast, no significant differences were found between cover crop treatments in AuT, with very similar total $N_{NO3}$ leaching amounts of approximately 20 mg N per core. These results indicate, firstly, that the volunteer treatment in NT significantly and considerably reduced N losses during the leaching experiment. Secondly, this effect was completely lost in the volunteer treatment of autumn-tilled samples. Importantly, there was also no significant difference between $N_{NO3}$ leaching between the bare fallow treatment in AuT and the bare fallow treatment in NT, indicating that tillage treatment alone did not reduce total N leaching from the intact cores.

Finally, all treatments have the same crop history, having been consistently kept under the same crop rotation since the establishment of the long-term trial in 2002. It is therefore highly unlikely that the reduction was caused by factors other than the establishment of volunteers and their interaction with tillage in the year of sampling.

**Table 1.** Nitrogen as $NO_3^-$ ($N_{NO3}$) leached from intact topsoil cores subjected to 200 mm of simulated rain. Mean values and 95% confidence intervals (CI) were obtained from linear mixed effects models with tillage and plant cover as main effects and experimental block as random effect.

| Tillage | Plant Cover | Mean Total Recovered $N_{NO3}$ (mg N) | 95% CI (mg N) | Group [1] |
|---------|-------------|--------------------------------------|---------------|-----------|
| NT | B | 18.4 | 12.09–24.62 | a |
| NT | V | 7.1 | 0.83–13.36 | b |
| AuT | B | 21.1 | 14.86–27.39 | a |
| AuT | V | 19.4 | 13.12–25.64 | a |

[1] Multiple pairwise comparisons significance threshold, *p* = 0.05.

As mentioned earlier, cover crop residence time is known to increase the overall effect of cover crops in reducing N leaching, with significantly lower N leaching with spring incorporation relative to autumn incorporation [10]. However, our results suggest that a difference in autumn cover crop residence of as little as one month can significantly hinder the efficacy of cover crops in reducing N leaching. Indeed, Sieling [17] highlight that N-uptake by cover crops is intrinsically linked to dry matter accumulation and therefore to intercepted photosynthetically active radiation. This means that cover crop residence is of particular importance for N uptake in the autumn when days are still relatively long (and warm). According to Vos and Van Der Putten [32], cover crops in autumn very broadly accumulate dry matter at a rate of 1.12 g per MJ intercepted global radiation. This figure allows us to make a broad calculation of the effect of autumn tillage on cover crops. Based on 65.5 MJ m$^{-2}$ total global radiation measured in November at Flakkebjerg station and an aboveground N content of 3.76% measured in plant clippings, the maximum potential N uptake by cover crops in AuT was 87 mg N per core lower than in NT. As discussed before, there are many other factors at play in determining the effect of cover crops in reducing N leaching after harvest, among them the fate of taken-up N once the cover crop is terminated. However, in terms of N uptake alone, prolonged field residence in early and mid-autumn is likely only second in importance to successful establishment of cover crops.

*3.2. Soil Nmin Content*

The mass of Nmin at the four dissected depths (Figure 3) was expectedly different in Reference and lab-rain cores, with clear depletion of soil mineral N after 200 mm of simulated rain in all treatments. Likewise, total mineral N contents in Reference and lab-rain cores show significant N losses during the leaching experiment in all tillage and plant cover treatments, except for NT-V (Table 2). These losses, approximately 22 mg N per core on average, closely resemble the total $N_{NO3}$ amounts recovered in the leachate for B in NT and both B and V in AuT.

Pairwise comparisons show that Nmin contents did not differ significantly between bare fallow and volunteer cores for either Reference or lab-rain cores in AuT. In contrast, pairwise comparison of NT Reference cores revealed a significantly higher total Nmin amount in B compared to V (13.6 mg N, SE = 4.96, *p* = 0.017) and, given that the volunteer cover crops were not incorporated in NT, this difference is attributable exclusively to a reduced cover crop N uptake. Interestingly, the amount of Nmin lost from NT-V cores was considerably smaller than in the AuT-B and AuT-V treatments. This supports our hypothesis that, through a combination of reduced uptake and re-mineralization, autumn tillage can hinder the intended function of cover crops.

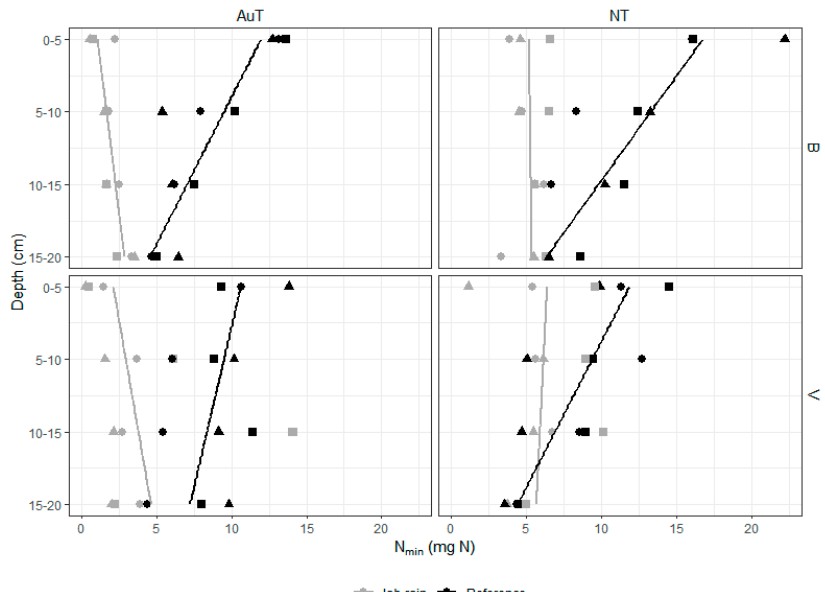

**Figure 3.** Mineral nitrogen (Nmin) amounts determined in four 5 cm layers in each intact core. Lab-rain cores underwent 200 mm of simulated rain in a laboratory lysimeter before dissection, while Reference cores were dissected without simulated rain. AuT and NT represent, respectively, inversion tillage carried out in late October and no-till. B and V represent bare fallow and winter rye volunteers as cover crops, respectively. The solid lines indicate linear interpolations of Nmin changes with depth.

**Table 2.** Total mineral nitrogen (Nmin) in intact cores dissected after 200 mm of simulated rain in laboratory lysimeters (lab-rain) and without simulated rain (Reference). AuT represents cover crop termination by inversion tillage approximately one month before sampling. NT represents no-till and continued cover crop residence until sampling. B and V represent bare fallow and winter rye volunteers as cover crops, respectively. Model estimates and 95% confidence intervals (CI) for Nmin were obtained from linear mixed effects models with experimental set tillage and plant cover as main effects and experimental block as random effect.

| Experimental Set | Tillage | Plant Cover | Total Nmin (mg N) | 95% CI (mg N) | Group [1] |
|---|---|---|---|---|---|
| Reference | | | 46.14 | 35.15–56.1 | a |
| Lab-rain | NT | B | 21.15 | 11.17–31.1 | b |
| Difference | | | 24.99 | | |
| Reference | | | 32.55 | 22.57–42.5 | a |
| Lab-rain | NT | V | 24.16 | 14.17–34.1 | a |
| Difference | | | 8.39 | | |
| Reference | | | 33.02 | 23.04–43.0 | a |
| Lab-rain | AuT | B | 7.98 | −2.01–18.0 | b |
| Difference | | | 25.05 | | |
| Reference | | | 35.75 | 25.76–45.7 | a |
| Lab-rain | AuT | V | 13.57 | 3.59–23.6 | b |
| Difference | | | 22.17 | | |

[1] Grouping by pairwise comparisons restricted to Reference and lab-rain treatments of the same tillage and plant cover treatments. Significance threshold, $p = 0.05$.

Unexpectedly, lab-rain NT cores retained on average approximately 5 mg N per dissected layer after the leaching experiment. In contrast, lab-rain AuT cores retained an average N mass per layer of 2 mg N in B and 3.4 mg N in V (Figure 3). The difference between AuT and NT lab-rain cores is likely due to a bypass effect, where soil solutes dispersed in the matrix are partially protected from leaching in soils experiencing heavy

precipitation. This effect has been found to be more prevalent in no-till and reduced till soils, as resident solutes in the matrix are increasingly bypassed by macropore water flow [33,34]. Importantly, this effect appears to be independent from cover crops treatment in our results, with no significant difference between B and V lab-rain total N content, suggesting that the soil N which is most strongly bypassed by macropore flow at high precipitation rates, is also poorly available for uptake by cover crops. Further research into the plant availability of matrix-associated Nmin in NT systems is necessary, particularly in the context of preferential exploration of pre-existing macropores by plant roots in more compact soils [35].

Organic elemental analysis of topsoil samples showed no significant main effects of cover crop or tillage on either total C or total N contents (mean values 15.13 mg C g$^{-1}$ and 1.37 mg N g$^{-1}$), in spite of near-significant trends of greater total C (3.6 mg C g$^{-1}$, SE = 1.58, $p$ = 0.052) and total N (0.29 mg N g$^{-1}$, SE = 0.149, $p$ = 0.088) in NT compared to AuT. The trend of greater total C content in NT resembles results by Gómez-Muñoz et al. [19], who found significantly higher soil total C contents in the upper 25 cm of NT soil in the same field experiment. However, lack of significant differences in our total C and N measurements and the fact that leaching was high in NT-B as well as AuT-B and AuT-V suggest that the results from the leaching experiment are not related to any underlying differences in total C and N soil contents. In Denmark, inversion tillage is typically carried out with a plough depth of 20–30 cm. It is therefore possible some volunteer plant material was buried below the sampling depth of the soil cores, in which case a portion of the corresponding re-mineralization of cover crop N would not have been captured by the leaching experiment. However, examination of the mineral N contained in the dissection layers at different depths (Figure 3) shows that Nmin is more evenly distributed across depths in AuT-V compared to AuT-B, suggesting that cover crop residues were mixed throughout the plough layer and the taken-up N had begun to re-mineralize in the one month span between tillage and sampling. Indeed, incubation studies have previously found that between 20% and 60% of the total N content in cover crop residues can mineralize within 1 month of incorporation at temperatures similar to those found in the field in autumn and winter [36,37].

### 3.3. Considerations on N Losses at Field Scale

Nutrient losses from the topsoil are of great relevance both for soil fertility and diffuse nutrient emissions from agriculture. Firstly, the root mass of many cash crops is strongly concentrated in the upper layers of the soil [38,39], and spring crops tend to proliferate new roots faster in the upper soil layers at early stages of development [40]. Secondly, the roots of many common cover crops, including winter rye, primarily explore the topsoil in the autumn [41], resulting often in greatest root growth in the upper 20 cm of the soil [42]. Thus, although autumn N leaching in the topsoil does not constitute a removal of said N from the soil column, it does reduce N availability for the following cash crop and increases the risk of diffuse N emissions into the environment.

Our breakthrough curve results indicate topsoil leaching losses of 2.2 kg N ha$^{-1}$ in NT-V and approximately 6.2 kg N ha$^{-1}$ in all other treatments, while soil core dissections show leaching losses of approximately 2.7 kg N ha$^{-1}$ and between 7 and 8 kg N ha$^{-1}$, respectively. These losses are much smaller than those commonly reported from field and outdoor lysimeter trials, which tend to average approximately 50 kg N ha$^{-1}$ and can be as high as 98 kg N ha$^{-1}$ (e.g., [11,12]). This is due primarily to the use of core samples rather than full soil columns, which limits the observations to the upper 20 cm of the soil. Additionally, it is likely some N was leached from the topsoil before sampling. The weather station at Flakkebjerg registered a total precipitation of 266 mm during the uncommonly wet 2019 autumn in Denmark [43], which would have carried significant amounts of NO$_3^-$ into the subsoil, below the sampling depth of our soil cores. Finally, it is also likely that some NO$_3^-$ was removed during core saturation and subsequent draining in the laboratory, although the total removal of N by drained water would have been limited as the volumetric water content of the cores was high after sample preparation (at a

pressure head of −10 hPa). In either case, given that all cores underwent the exact same sample preparation procedure regardless of treatment, the differences between treatments (or lack thereof) remain informative in spite of the reduced total leaching amounts.

We acknowledge the important distinction between N leached from the intact cores in this study, and N leaching in the field. Extrapolating mesocosm and laboratory results to the field scale is not entirely straightforward, given the natural variability of the soil and the limited representation of this variability that a small sample can provide. However, Valkama et al. [12] found no significant differences in results from field and lysimeter analyses in a meta-analysis of N leaching losses that included 13 field experiments and 6 lysimeter experiments. Furthermore, the representative quality of other measurements, e.g., suction cup or tile drain measurements, has also been questioned as soil N content and actual water drainage in the field remain difficult to determine [44], forcing studies to extrapolate or model some part of their results. Thus, exploiting the increased opportunities for controlled drainage and soil analysis afforded by laboratory-scale analyses remains valuable in the study of N leaching in agriculture.

**Author Contributions:** Conceptualization, J.F.M.-V. and I.V.; methodology, J.F.M.-V. and I.V.; validation, I.V.; formal analysis, J.F.M.-V.; investigation, J.F.M.-V. and I.V.; data curation, J.F.M.-V.; writing—original draft preparation, J.F.M.-V.; writing—review and editing, I.V.; visualization, J.F.M.-V.; supervision, I.V.; project administration, I.V.; funding acquisition, I.V. All authors have read and agreed to the published version of the manuscript.

**Funding:** This research was funded by the Aarhus University Research Foundation (AUFF) Starting Grant.

**Institutional Review Board Statement:** Not applicable.

**Informed Consent Statement:** Not applicable.

**Data Availability Statement:** The data analyzed and reported in this study is openly available in FigShare at doi:10.6084/m9.figshare.19354721.

**Acknowledgments:** We would like to thank Lars Juhl Munkholm and Elly Møller Hansen for their help in the execution of this project, and for granting access to the long-term reduced tillage experiment at Aarhus University Flakkebjerg. Additionally, we would like to thank Michael Koppelgaard, Stig Rassmussen, Karen B. Heinager and Eugene Driessen for their technical assistance in the field and laboratory.

**Conflicts of Interest:** The authors declare no conflict of interest.

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
