# Peer review of "Autumn Tillage Reduces the Effect of Plant Cover on Topsoil Nitrogen Leaching"

_nitrogen, doi:10.3390/nitrogen3020014_

Round 1
Reviewer 1 Report
Dear Author,
This manuscript contributes several observations about the nitrogen leaching process. However, there are some issues that can be improved in the paper "Autumn tillage reduces the effect of crop cover on nitrogen leaching from topsoil":
- Please consider including Latin names for plants in the rotation
- correct references in the text to [1], [1-3] or [1,3]
- correct units, e.g. L/ha to L ha-1 etc.
- Normally, soil nitrogen transformations are correlated with changes in SOM. Some effects observed in the experiment are clearly related to organic matter. Was the SOM content constant throughout the experiment?
- Basic soil physicochemical parameters were insufficiently described. Why initial N content was not considered, but only texture, density, SOM content?
- What about changes in P and Ca during the experiment? These elements strongly affect all processes concerning leaching/accumulation of mineral nitrogen...
- In studies of soil nutrient leaching processes, it is obligatory to measure changes in pH of both soil and leachate.
In conclusion, the reviewed paper is interesting but needs major revision to be published.
Reviewer 2 Report
The manuscript titled “Autumn tillage reduces the effect of plant cover on topsoil nitrogen leaching” is an interesting laboratory-scale work on soil N movement under different tillage and soil cover conditions. Although the variability is large and the number of cores is small, the results obtained, may be of interest to increase the knowledge about the management of cover crops in northern European areas. The manuscript is well organized and the different sections reflect well the work done. Moreover, the manuscript presents the results of the analysis in a logical manner. The subject falls within the scope of the Nitrogen, and might be of interest for its readership. Specific comment and suggestions to the authors are included in order to improve the final version of the manuscript.
General comments:
- It would have been interesting to know also the distribution of ammonium and nitrate in the soil cores, not only Nmin. Also if ammonium leaching occurred.
Specific remarks:
- Line 16: Change CT to AuT
- Line 120: Explain why such a dilution is used in the simulated rainwater experiment.
- Line 122: It is not clear from the text whether the total simulated precipitation (200 mm) is applied at one time or at different time periods.
- Figure 2 and others: Any comments on the greater variability of the results obtained in the no-tillage treatments?
- Table 1 and 2: …95% confidence intervals (CI) were…
- Lines 254 – 255: “… the Nmin lost from NT-V cores was 13.8 and 16.7 mg N lower than in the AuT-B and AuT-V treatments, respectively…” Are the values changed in the treatments?
Round 2
Reviewer 1 Report
Dear authors, Thank you for including most of the suggested corrections. Therefore, I accept this manuscript for publication in Nitrogen.
Regarding comment 6: please note that both P and N are mainly leached in anionic form and the mechanism and intensity of this process is quite different than in the case of cations.